# OpenReview forum: "MIRROR: Modular Internal Processing for Personalized Safety in Multi-turn Dialogue"
_ICLR.cc/2026/Conference — ICLR 2026 Conference Withdrawn Submission_

### Official Review · Reviewer_3Pro · 2025-10-30

**Soundness:** 3
**Presentation:** 4
**Contribution:** 3
**Rating:** 6
**Confidence:** 3

**Summary:**

This paper proposes a two-component framework called MIRROR for personalized safety assurance in multi-turn dialogues. The framework separates user-facing rapid response (Talker) from asynchronous reasoning (Thinker), where the Thinker maintains a persistent, limited, and self-updating internal state across turns. The Thinker includes an internal monologue manager that generates parallel threads (goals, reasoning, memory), and a cognitive controller that synthesizes these threads into a concise internal narrative in each dialogue turn. The Talker then utilizes this narrative to maintain safety awareness without exposing thought processes or adding blocking latency.

In the CuRaTe benchmark (1685 dialogues across five scenarios), MIRROR improved the average success rate of seven models from 69% to 84%, with open-source models Llama 4 Scout (91%) and Mistral Medium 3 (90%) outperforming the GPT-4o and Claude 3.7 Sonnet baselines, while adding only minimal per-turn overhead to the open-source models and an average of approximately 460 milliseconds of latency in a production-like setting.
This architectural design enhancements can address critical personal safety failures (context drift, conformity bias, sycophancy) across multi-turn dialogue.

**Strengths:**

This paper is well-motivated , and the authors point out that existing systems cannot simultaneously satisfy all the desired characteristics. Their work proposes a secure multi-turn dialogue method suitable for production environments.

They evaluated seven different LLM models, reported their performance in different scenarios, and provided a detailed analysis of the costs and latency under limited computing resources, which is a very practical consideration.

The paper is clearly written, with well-structured diagrams, and provides detailed descriptions of the prompts and state management details.

**Weaknesses:**

the method's behavior may be sensitive to prompts, but robust ablation studies are currently lacking.(e.g., performance degradation of GPT-4o in Scenario 3, Table 10)

The evaluation is performed only with 5-turn dialogues (1 introduction + 3 distractors + 1 critical question) in CuRaTe benchmark. Its generalization ability and robustness beyond a five-turn conversation structure are uncertain.

No empirical comparison of related reflection/memory systems (e.g., Reflexion, LATS, Devil’s Advocate, Sleep-Time Agents, etc.) on the same benchmark. Even if not all related work has all properties that MIRROR has, from my understanding they're addressing the same task, so it would be good to show some performance comparison, or even just microbenchmark.

**Questions:**

Could you implement or adapt at least one baseline on CuRaTe to provide a comparison, reporting both safety and latency so we can see Pareto trade-offs?​

How does performance degrade as conversations lengthen beyond 5 turns?    can you add a long-dialogue stress test?

---

### Official Review · Reviewer_5iaz · 2025-10-31

**Soundness:** 2
**Presentation:** 3
**Contribution:** 2
**Rating:** 2
**Confidence:** 3

**Summary:**

The paper introduces MIRROR, a dual-component architecture designed to solve the critical failure of LLMs forgetting user-specific safety constraints (like allergies or trauma) in multi-turn dialogue. Inspired by Dual Process Theory, it uses a "Talker" for immediate responses and an asynchronous "Thinker" that performs background deliberative reasoning.

**Strengths:**

1. **Targets a Specific Problem**: The architecture addresses a documented failure where LLMs forget user safety constraints when conversations drift. This is a well-defined engineering problem.
2. **Separates Speed from Thinking**: The Talker/Thinker design splits instant responses from heavy computation. The Thinker runs asynchronously during natural conversation pauses to hide latency. Smart use of idle time.
3. **Shows Measurable Gains**: On the CuRaTe benchmark testing this exact problem, MIRROR models improved from 69% to 84% average performance.
4. **Maintains Constant Cost**: Using a bounded 3k token state that regenerates each turn keeps overhead at O(1) instead of O(n) growth. Theoretically sound design, though whether 3k is enough needs verification.

**Weaknesses:**

1. **Single Benchmark Testing**: All performance claims rely on one benchmark, CuRaTe. Need experiments on other benchmarks to verify the safety improvements don't harm performance on general tasks.
2. **Missing Standard Baselines**: The paper compares against vanilla LLMs but skips simpler solutions like RAG or long context windows. These alternatives might actually be superior (longer context retention, potentially lower cost). Without comparative experiments, the necessity of MIRROR's complexity remains questionable.
3. **First Turn Vulnerability**: By design, the Talker has no internal state at t=0. If a user states their constraint and asks a dangerous question in the same initial message, they get an unprotected baseline response.
4. **Incomplete Cost-Benefit Analysis**: Needs deeper analysis comparing cost-effectiveness against simpler alternatives across different use cases
5. **State Lag Problem**: Talker at turn t uses state from turn t-1. If user intent reverses between turns (wants dangerous thing → actually no), the response uses outdated context. Potential for contradictory advice.
6. **Lossy Compression Risk**: The Cognitive Controller must compress full history into 3k tokens by discarding information. If it drops critical safety data (old allergy for new trauma), that information is permanently gone.
7. **Ablation Contradictions**: Table 3 shows Full MIRROR (82%) performs worse than Cognitive Only (87%) on Claude 3.7 Sonnet. This suggests the full architecture may degrade performance on advanced models. Requires more detailed discussion of when and why different components help or hurt.

**Questions:**

How does performance degrade with multiple simultaneous safety constraints? What about conversations longer than the 5-turn benchmark scenarios?

How does MIRROR compare against simpler alternatives like RAG or long context windows in terms of both performance and cost? What specific advantages justify the added architectural complexity?

What is the actual cost-performance tradeoff across different models and use cases? At what conversation length or constraint complexity does MIRROR become more economical than alternatives?

What information does the Cognitive Controller prioritize when compressing to 3k tokens? How often does critical safety information get lost, and can this be quantified?

---

### Official Review · Reviewer_raSv · 2025-11-01

**Soundness:** 2
**Presentation:** 2
**Contribution:** 1
**Rating:** 2
**Confidence:** 4

**Summary:**

This paper presents MIRROR, a production-oriented dual-process architecture that decouples the immediate, low-latency responder (Talker) from an inter-deliberative component (Thinker) while maintaining a persistent, bounded, regenerative internal state to carry user-specific safety constraints across turns. On the CuRaTe benchmark, the authors report large safety gains across 7 models.

**Strengths:**

- The paper is generally well written with scoped failure modes (context drift, sycophancy, conformity bias) as structure, with a proper benchmark as experiment setup.

- The paper’s inter-turn orchestration with a bounded, regenerative internal state and a two-stage Thinker is a coherent systems contribution aimed at benefitting personalized safety.

- This paper also considers deployment framing with latency and cost analyses;

**Weaknesses:**

- The claim all rely on a single benchmark; a single bespoke benchmark is not enough for strong claims with potential overfitting to single task and metric.

- The paper repeatedly quotes success rates and large deltas, but does not clearly define the labeling rubric, judge setup (human vs LM),  or uncertainty. The results also lack error bars or variance across seeds.

- Baselines are chosen not clearly positioned against state-of-the-art inter-turn reflection/memory agents that also maintain bounded, regenerative summaries or include safety critics.

- The cost analysis, which relies on OpenRouter, gives prices and a maximum token overhead; many of the claims like "fundamentally inverts the economics of AI safety." are over-stated.

- It seems most of the proposed novelty is conceptually adjacent to reflection/memory or as in agents like inner monologue, summaries, safety reminders. I don't see huge novelty in packaging this into conversations.

**Questions:**

I wonder if the authors have compared MIRROR with other similar works on their performance?

Also instead of latency analysis, more ablation studies could help, e.g. on different state size or thinker setting could help understand the performance improvements.

---

### Official Review · Reviewer_zVPv · 2025-11-01

**Soundness:** 2
**Presentation:** 2
**Contribution:** 2
**Rating:** 0
**Confidence:** 4

**Summary:**

To address the issue that LLMs often generate harmful responses while ignoring user-specific safety constraints, this paper proposes a modular production-focused architecture to prevent these failures. Specifically, based on Dual Process Theory, this paper proposes an LLM system consisting of a Thinker and a Talker, in which the former one focuses on deep and asynchronous consolidation and the latter generates an immediate response based on the internal state updated by the former one. The experiments are conducted on a personalized safety benchmark and achieve superior performance over some open-sourced and proprietary LLMs.

**Strengths:**

1. It proposes a modular deployment-time architecture consisting of a Thinker to consolidate the reasoning process and contextual history into an internal state and a Talker to generate immediate responses based on the continuously updated internal state. This architecture is motivated by a theory of human cognition.

2. It increases overall safety performance across multiple models in a production-like environment using a smaller open-source model with minimal additional cost.

**Weaknesses:**

1. This paper is more engineering-oriented. Though it achieves superior performance across several models on an LLM safety benchmark, it seems to contain little new algorithms or architectures. The dual system (fast-and-slow) is widely used across many areas, including but not limited to machine learning, computer vision, and robotics.

2. Another concern is implementation. In Figures 2 to 4, each component or module is illustrated to achieve several objectives. For example, the Cognition Controller needs to "synthesize ..., consolidate ..., plan ...", but I am concerned whether these objectives can be really achieved using a single API/function call, as mentioned in Line 188. This is just one example.

These are the two main but also fundamental concerns regarding this paper.

**Questions:**

Please refer to the weaknesses section.

---

### Note · Authors · 2025-11-21

**Comment:**

We thank all reviewers for their time and thoughtful engagement with our work. We are withdrawing this submission to strengthen it and appreciate the feedback provided.

**Withdrawal Confirmation:**

I have read and agree with the venue's withdrawal policy on behalf of myself and my co-authors.